# Management of COVID-19-Associated Acute Respiratory Failure with Alternatives to Invasive Mechanical Ventilation: High-Flow Oxygen, Continuous Positive Airway Pressure, and Noninvasive Ventilation

**DOI:** 10.3390/diagnostics11122259

**Published:** 2021-12-02

**Authors:** Barbara Bonnesen, Jens-Ulrik Stæhr Jensen, Klaus Nielsen Jeschke, Alexander G. Mathioudakis, Alexandru Corlateanu, Ejvind Frausing Hansen, Ulla Møller Weinreich, Ole Hilberg, Pradeesh Sivapalan

**Affiliations:** 1Department of Medicine, Section of Respiratory Medicine, Herlev and Gentofte Hospital, University of Copenhagen, 2200 Copenhagen, Denmark; barbara.bertelsen@regionh.dk (B.B.); jens.ulrik.jensen@regionh.dk (J.-U.S.J.); 2Department of Clinical Medicine, Faculty of Health Sciences, University of Copenhagen, 2200 Copenhagen, Denmark; 3Department of Respiratory Medicine, Copenhagen University Hospital-Hvidovre, 2650 Hvidovre, Denmark; klausnielsen.md@gmail.com (K.N.J.); ejvind.frausing.hansen@regionh.dk (E.F.H.); 4Division of Infection, Immunity and Respiratory Medicine, School of Biological Sciences, The University of Manchester, Manchester Academic Health Science Centre, Manchester M23 9LT, UK; Alexander.Mathioudakis@Manchester.ac.uk; 5North West Lung Centre, Wythenshawe Hospital, Manchester University NHS Foundation Trust, Manchester M23 9LT, UK; 6Department of Respiratory Medicine, State University of Medicine and Pharmacy “Nicolae Testemitanu”, 2004 Chisinau, Moldova; alexandru.corlateanu@usmf.md; 7Department of Respiratory Medicine, Aalborg University Hospital, University of Aalborg, 9100 Aalborg, Denmark; ulw@rn.dk; 8The Clinical Institute, Aalborg University, 9220 Aalborg, Denmark; 9Department of Medicine, Little Belt Hospital, 7100 Vejle, Denmark; ole.hilberg@rsyd.dk; 10Department of Regional Health Research, University of Southern Denmark, 5000 Odense, Denmark

**Keywords:** COVID-19, acute respiratory failure, noninvasive ventilation, NIV, bilevel positive airway pressure, BiPAP, continuous positive airway pressure, CPAP, high-flow nasal cannula oxygen therapy, HFNC

## Abstract

Patients admitted to hospital with coronavirus disease 2019 (COVID-19) may develop acute respiratory failure (ARF) with compromised gas exchange. These patients require oxygen and possibly ventilatory support, which can be delivered via different devices. Initially, oxygen therapy will often be administered through a conventional binasal oxygen catheter or air-entrainment mask. However, when higher rates of oxygen flow are needed, patients are often stepped up to high-flow nasal cannula oxygen therapy (HFNC), continuous positive airway pressure (CPAP), bilevel positive airway pressure (BiPAP), or invasive mechanical ventilation (IMV). BiPAP, CPAP, and HFNC may be beneficial alternatives to IMV for COVID-19-associated ARF. Current evidence suggests that when nasal catheter oxygen therapy is insufficient for adequate oxygenation of patients with COVID-19-associated ARF, CPAP should be provided for prolonged periods. Subsequent escalation to IMV may be implemented if necessary.

## 1. Introduction

Patients infected with severe acute respiratory syndrome coronavirus 2 (SARS-CoV-2) may develop coronavirus disease 2019 (COVID-19) with viral pneumonia, acute respiratory failure (ARF), or acute respiratory distress syndrome (ARDS) and may require hospital admission [1,2,3]. ARF is defined as the inability of the respiratory system to meet the oxygenation demands, ventilation, or metabolic requirements of the patient [4]. Treating patients with COVID-19 who have ARF involves oxygen supplementation and, in some cases, a degree of assisted ventilation. In the most severe cases of hypoxemia, invasive mechanical ventilation (IMV) may be necessary. However, access to IMV therapy may be limited, and this should be reserved for cases in which it is clearly indicated. IMV can result in complications linked to the intubation procedure and increased risks of ventilator-induced lung injury and ventilator-associated pneumonia, as well as long-term complications such as critical illness polyneuromyopathy [5,6,7]. Consequently, many intensive care units (ICUs) and nonintensive care medical departments looked for alternatives to IMV during the initial surge in COVID-19 cases. These alternatives included bilevel positive airway pressure (BiPAP), continuous positive airway pressure (CPAP), and high-flow nasal cannula oxygen therapy (HFNC). One substantial benefit of IMV is that it operates within a closed system, resulting in minimal spread of viral particles. Some clinicians were reluctant to use BiPAP, CPAP, and HFNC during the initial phase of the COVID-19 pandemic, due to the potential risk of transmission to healthcare staff [8,9,10]. The different types of ventilation treatment are associated with different risks of particle dispersion and disease transmission. Studies have shown that CPAP is not associated with significant leakage of exhaled air, whereas a single BiPAP circuit resulted in exhaled air reaching a distance up to 0.92 m from the BiPAP apparatus [11]. Similarly, a double BiPAP circuit was not associated with significant leakage. It may be relevant to compare the associated risks when ARF treatment efficacies are similar; however, as a German study stated, fear of transmission must not become the basis for selecting which ventilation method to use for patients [12]. Therefore, determining the best treatment options and alternative ventilation methods for patients with COVID-19 is critical. This review investigates the current treatment of patients with COVID-19 and ARF, as well as relevant ventilation strategies.

## 2. Acute Respiratory Failure and COVID-19

ARF can be categorized as type I or type II. Type I ARF is characterized by hypoxemia with a reduced partial pressure of oxygen in arterial blood (PaO_2_). This is the type of respiratory failure most frequently observed in patients with COVID-19 who have had no prior respiratory illness or have had low levels of exposure to tobacco smoke [2]. Type II ARF is characterized by hypercapnia with an increased partial pressure of CO_2_ in arterial blood (PaCO_2_) [13]. Patients with hypoxemic (type I) and hypercapnic (type II) ARF may benefit from different oxygenation strategies, to minimize the risk of deterioration and the requirement for IMV [14]. Evidence is accumulating that the course of hypoxemic lung injury in COVID-19 pneumonia may be more heterogeneous and may differ in various ways from the course of the disease in other pathogenic contexts [15,16]. In COVID-19 pneumonia, hypoxemic lung injury is accompanied by damage to the vascular endothelium and an increased risk of multiple organ failure. Therefore, COVID-19 pneumonia can be viewed as a systemic disease [12].

## 3. Oxygenation Targets for Patients with COVID-19 and Respiratory Symptoms

The HOT ICU trial, a randomized controlled trial (RCT) of 2928 patients, demonstrated that patients with ARF do not benefit from a target PaO_2_ of 60 mm Hg in comparison to target PaO_2_ of 90 mm Hg [17]. A smaller RCT found that in patients with ARDS but no exposure to SARS-CoV-2, liberal oxygen therapy (i.e., targeting peripheral oxygen saturation (SpO_2_) of >96%) did not increase survival at 28 days, compared with conservative oxygen therapy (i.e., targeting SpO_2_ of 88–92%) [18]. The British Thoracic Society recommends that oxygen should be prescribed to achieve a target saturation of 94–98% for most acutely ill patients, with a patient-specific target saturation of 88–92% for patients at risk of type II (hypercapnic) respiratory failure [19], and the Surviving Sepsis Campaign guidelines recommend a target SpO_2_ of 92–96% [20]. However, oxygenation goals for patients with severe illness and respiratory symptoms should always depend intrinsically on underlying factors, and liberal oxygen treatment may increase the risk of mortality in patients with acute cerebral or coronary ischemia [21], as well as in those in ICUs [22].

The World Health Organization recommends a target SpO_2_ of ≥90% for nonpregnant (≥92–95% for pregnant) patients with COVID-19-associated ARF and also recommends reaching these targets by titration via a nasal cannula, simple face mask, or a face mask with a reservoir bag [23] (Figure 1). Studies on optimal target oxygenation in patients with COVID-19-associated ARF are scarce, and no RCTs have been performed to the best of our knowledge. An additional challenge on this subject is the traditional mode of prescribing oxygen therapy, which is often not documented or specified to the degree, which is common with pharmaceutical therapies [24]. Patients with COVID-19-associated ARF, despite the provision of maximal oxygen levels via a face mask, should be promptly identified and evaluated with a view to providing respiratory support via HFNC, CPAP, or intubation and mechanical ventilation [25] (Figure 1).

## 4. Diagnosing Acute Respiratory Failure

In hypoxemic (type I) respiratory failure patients are diagnosed by a peripheral saturation measurement or preferably an arterial gas analysis. Hypoxemic respiratory failure is defined as a lowered PaO_2_ or SpO_2_ measurement (hypoxemia) with either a standard (normocapnia) or low (hypocapnia) PaCO_2_ measurement. For a certain diagnosis of respiratory failure, PaO_2_ < 60 mmHg by arterial gas analysis is mandatory.

In patients with COVID-19-associated ARF, the PaO_2_/FiO_2_ ratio was able to predict severity in a cohort of 421 patients [26], as well as in a smaller cohort of 150 patients. In the latter cohort, the authors were able to calculate an optimal cut-off PaO_2_/FiO_2_ ratio of 274 mmHg, which could distinguish between patients with mild disease and patients with severe disease with a sensitivity of 72% and specificity 85% [27], and it also contributed to a predictive composite score along with age, platelets, pH, blood urea nitrogen, temperature, PaCO_2_, and Glasgow Coma Scale in a cohort of 937 patients [28].

A study has examined an algorithm for predicting COVID-19-associated ARDS among 964 patients who would develop ARDS within 12 h, as compared to 4712 patients who did not. The machine algorithm found that the two most important predictive factors for the development of COVID-associated ARDS were saturation in the shape of the lowest measured SpO_2_ and standard deviation of measured SpO_2_. This was complemented by age, maximal FiO_2_, maximal respiratory rate, and maximal and standard deviation O_2_ flow. Hence, and perhaps not surprising, several different respiratory measures seem to contribute substantially to predicting ARDS in patients with COVID-19. However, as a possible future biomarker, the lowest measured platelet count also contributed to the algorithm [29]

In patients without COVID-19, several biomarkers seem promising in diagnosing and predicting the development of ARF, and some have already to some degree proven their value. Plasma surfactant protein D (SP-D) has been shown to increase within 48 h of admission to ICU in patients who developed ARDS and to predict the long-term need for IMV and mortality [30,31,32], and angiopoietin-2 was able to predict pulmonary affection in cohort studies in critically ill patients with various underlying courses [33,34,35,36] and also predict severity of illness and mortality [37,38,39,40].

Among patients with COVID-19, a cohort study of 259 patients found that SOFA score and ROX index predicted HFNC failure to IMV [41], and in a small cohort, male sex, obesity, African American ethnicity, comorbidities, and prior immunosuppression predicted HFNC failure and need for IMV [42], though in this small study, no biomarker was able to predict a respiratory worsening [42]. A retrospective cohort study of 610 patients showed a compound of age, history of coronary heart disease, CRP, aspartate aminotransferase, D-dimer, and neutrophil-to-lymphocyte ratio was able to form an acceptable ROC curve [43], and in a retrospective cohort study of 638 patients, CRP, neutrophil-to-lymphocyte ratio, and D-dimer were associated with adverse events, such as acute myocardial injury, respiratory failure, acute kidney injury, mechanical ventilation, intensive care unit admission, multiple organ dysfunction syndromes, and death [44]. 

Elevated levels of LDH have also been associated with severity [45,46], and in a cohort study of 100 patients, ferritin was able to predict in-hospital mortality with a superior ROC curve to CRP [47], and in a cohort of 153 patients, severity of disease was associated with plasma levels of interleukin-6 (IL-6), CRP, soluble-IL2 receptor (IL2Rα), procalcitonin (PCT), and ferritin [48]. IL-6 has also been shown to correlate to SpO_2_ and PaO_2_ in 48 patients, and it correlated well with CRP in these patients [49]. This is not in contrast to studies on patients with ARF without COVID-19; however, in this context, IL-6 was a somewhat inconsistent predictor of ARF, ARDS, severity, and mortality [37,38,50,51,52]. Similarly, PCT has in patients without COVID-19 been a poor indicator for need for antibiotic treatment [53].

## 5. Continuous Positive Airway Pressure

CPAP is a positive airway pressure therapy that delivers a set pressure of airflow to the airways. The C for continuous in CPAP refers to the constant pressure, which is maintained throughout the respiratory cycle, both when the person is breathing in (inspiration) and breathing out (expiration). This therapy can be administered for prolonged periods of time, referred to as continuously administered CPAP (cCPAP), or for very short time intervals, referred to as intermittently administered CPAP. In this review, we only discuss cCPAP therapy. 

A CPAP device consists of a unit that generates airflow, which is delivered to the airway through either a helmet or face mask, and the effects of CPAP have been studied in more than 1100 patients with ARF due to COVID-19 pneumonia [3]. Continuous therapy with CPAP for COVID-19-associated ARF may be considered when a patient with hypoxic respiratory failure requires 6–15 L/min oxygen (i.e., fraction of inspired oxygen (FiO_2_), 0.4–0.6) to achieve an acceptable level of oxygen saturation, and clinicians agree that escalation to IMV would be an option, but it is not immediately necessary (Figure 1). When commencing the CPAP therapy, an initial pressure of 10–12 cm H_2_O may be applied, because positive end-expiratory pressure should be high, as in other cases of severe ARF [25,54,55]. Usually, CPAP therapy has an almost instantaneous effect on improving the condition of patients with COVID-19-associated ARF; however, if more oxygen is necessary, then intubation and IMV may be required [25]. 

The RECOVERY-RS trial, led by the University of Warwick and Queen’s University Belfast, is the world’s largest noninvasive respiratory support trial for COVID-19, with over 1200 participants taking place across 48 United Kingdom hospitals. This multicenter adaptive RCT compared the use of CPAP (oxygen and positive pressure delivered via a tightly fitting mask), HFNC, and standard care (standard oxygen therapy). The results showed that treating hospitalized COVID-19 patients who had ARF with continuous CPAP reduced the need for IMV [56]. In the CPAP group, 137 of 377 participants (36.3%) either needed mechanical ventilation or died within 30 days, compared with 158 of 356 participants (44.4%) in the conventional oxygen therapy group (unadjusted odds ratio, 0.72; 95% confidence interval (CI), 0.53 to 0.96; *p* = 0.03). However, there was a small increase in the number of serious adverse events with CPAP compared with conventional oxygen therapy [56]. On the other hand, one small RCT and several cohort studies have shown that continuous CPAP therapy is not successful for all patients, and failure rates are higher when the treatment ceiling excludes IMV, such as when treating elderly patients who have many comorbidities [55,57,58,59,60,61,62]. Continuous CPAP has particularly beneficial effects on respiratory rate and oxygenation levels, and the majority of patients with COVID-19 who are offered continuous CPAP therapy (83–97%) are able to tolerate the treatment [57,63]. CPAP weaning may be undertaken reasonably safely (83% success rate) after treatment has successfully improved respiratory performance and the modified ROX index (PaO_2_/FiO_2_/respiratory rate) has increased to 8.4 mm Hg/bpm [62]. Initial 6-month follow-up studies of survivors of COVID-19-associated ARF treated with continuous CPAP were optimistic concerning respiratory parameters and exercise capacity [64].

## 6. Bilevel Positive Airway Pressure

BiPAP ventilates by applying positive pressure into the lungs through a mask or a helmet [38]. BiPAP can be used as initial treatment, followed by a step up to IMV if needed, or as a method for weaning patients off IMV. BiPAP is very effective, and many guidelines describe BiPAP as the first-line treatment for ARF caused by acute exacerbations of chronic obstructive pulmonary disease or acute cardiogenic pulmonary edema [65,66]. One study investigated the efficacy of BiPAP in treating ARF resulting from various etiologies [67]. The highest BiPAP failure rate was among patients with hypoxemic respiratory failure, and the lowest was among patients with acute pulmonary edema [41]. Previous studies have shown that BiPAP can effectively treat viral pneumonia with hypoxic respiratory failure. Failure rates were as low as approximately 30%. For influenza A (H1N1), the failure rate was 13–77% [66]. Potential difficulties associated with BiPAP treatment include low patient compliance, claustrophobia, dyspnea, and development of skin rashes [6,68]. 

BiPAP may be used to treat COVID-19-associated ARF at an initial inspiratory pressure of 20 cm H_2_O and an expiratory pressure of 10 cm H_2_O combined with low tidal volumes such as 4–8 mL/kg [25,54]. The HENIVOT trial, a multicenter RCT of 110 patients (median age, 65 years), evaluated whether helmet BiPAP for two days followed by HFNC therapy was superior to HFNC alone. Though the primary outcome was not met, among the secondary outcomes, the rate of endotracheal intubation was significantly lower in the BiPAP group than in the HFNC group (30% vs. 51%; *p* = 0.03). These data are obviously not conclusive, and the area needs further investigation [69]. As with CPAP treatment, age, comorbidities, and the severity of the disease predicted BiPAP treatment failure and mortality in patients with COVID-19-associated ARF [70,71]. Treating patients who are in the prone position with BiPAP has produced promising results; however, the relevant data are limited because few patients are initially treated in the prone position [72,73]. Overall, BiPAP failure rates ranged from 48 to 53% [70,71]. Finally, BiPAP has been used as a step-down measure from IMV, with BiPAP being implemented immediately after early extubation in patients with severe COVID-19-associated ARF. This strategy reduced the duration of intubation, the extubation failure rate, and the need for reintubation. However, patients treated with BiPAP had also been given more antiviral agents and more corticosteroids. Larger studies are needed to verify the few preliminary data on these BiPAP strategies. Both CPAP and BiPAP treatment are associated with a considerable risk of complications such as pneumothorax/pneumomediastinum, hemodynamic instability, and delirium and require careful monitoring [56,57,74]. These risks may be increased among patients suffering from COVID-19-associated ARF [75].

## 7. High-Flow Nasal Cannula Oxygen Therapy

HFNC involves the delivery of a high flow of warm humidified oxygen (up to 60 L/min) through a small nasal cannula, improving oxygenation and reducing respiratory rates, as well as providing higher concentrations of oxygen than therapy with supplemental oxygen alone [76,77]. In patients who did not have COVID-19, the European Respiratory Society (ERS) recommends HFNC therapy to patients with hypoxic respiratory failure over conventional nasal cannula therapy and NIV; however, they do not recommend HFNC over NIV in patients at high risk of extubation failure, in patients with chronic obstructive pulmonary disease (COPD), or in patients with hypercapnic ARF [78]. Treatment with HFNC resulted in similar mortality rates but less frequent intubation, compared to patients treated with conventional oxygen therapy [79], and HFNC is usually tolerated quite well [80]. 

Preliminary data from a recent RCT involving HFNC treatment of COVID-19 patients with ARF (*n* = 1272) suggested that routine use of HFNC did not improve patient outcomes, compared with conventional oxygen therapy. The RCT did not find any benefits associated with using HFNC rather than conventional oxygen therapy. A total of 184 of 414 participants who received HFNC (44.4%) vs. 166 of 368 participants who received conventional oxygen therapy (45.1%) met the composite endpoint of tracheal intubation or death within 30 days (unadjusted odds ratio, 0.97; 95% CI, 0.73 to 1.29; *p* = 0.85), whereas continuous CPAP was superior to conventional oxygen therapy [56]. Therefore, routinely offering HFNC as the main form of respiratory support for patients with respiratory failure due to COVID-19 is not recommended [81], but HFNC may be suitable for patients who need a break from CPAP (e.g., at mealtimes) or for patients who are being weaned off CPAP or need humidified oxygen or for patients who cannot tolerate CPAP. As described above, evidence from the HENIVOT trial [69] suggests that for COVID-19 patients with moderate-to-severe hypoxemia, treatment with helmet noninvasive ventilation did not increase the number of days free of respiratory support within 28 days, compared with high-flow nasal oxygen (mean difference, 2 days; 95% CI, –2 to 6; *p* = 0.26). In addition, helmet noninvasive ventilation followed by HFNC significantly reduced the number of patients who needed invasive ventilation, compared with HFNC alone. Previous small national studies on clinical outcomes in patients with COVID-19 who were treated with HFNC were mostly uncontrolled and retrospective and did not reach definitive conclusions [82,83,84]. This may be partly due to confounding variables and biases, as well as the difficulties associated with extrapolating results from one population to another. In addition, some of these studies primarily had exploratory relevance in the earlier stages of the SARS-CoV-2 pandemic. Looking at the use of HFNC from another point of view, a retrospective study of patients with COVID-19 who developed ARF found that 41% of those treated with HFNC experienced treatment failure and required BiPAP or intubation. The HFNC treatment failure rate was highest among patients with low PaO_2_/FiO_2_ ratios. This marker is frequently used in intensive care settings and may help to identify those patients with COVID-19 and ARF who are most likely to require escalation in therapy from HFNC [85]. Further research is needed to clarify particular aspects of HFNC, including treatment targets, safety, efficacy, and how it should be administered to patients with ARF and COVID-19.

## 8. Prone Positioning

Positioning as a therapy method in non-ICU patients with COVID-19-associated ARF has also been examined to better oxygenate and avoid IMV. So far, studies on positioning in non-ICU patients have been conducted with prone positioning undertaken only during waken periods. Two small RCTs have been conducted so far. An RCT feasibility study of 60 patients with hypoxic ARF secondary to COVID-19 pneumonia found no effect of prone position on the need for additional respiratory therapy or mortality; however, only 13 patients encouraged to lay in the prone position were able to self-prone for at least 6 h a day. In the standard care group, 16 patients chose by themselves to spend time in the prone position, which may have masked an effect [86]. A pilot trial designed as a cluster study in a quaternary care center included five inpatient medical service teams to either encourage prone positioning or standard care with no randomization of patients found no effect on oxygen saturation to fraction of inspired oxygen ratio of encouragement of medical service teams to prone positioning [87]. Finally, a small cohort study of 20 patients examined peripheral oxygen saturation and showed an improvement from 96% in the supine position to 98% in the prone position (*p* = 0.008). However, the patients reported worsening in comfort score [88]. Hence, any effect of prone positioning in non-ICU patients remains to be shown but can also not be rejected with the currently available evidence.

## 9. Stepping up the Therapeutic Regimens

A retrospective observational study [89] investigated moderate-to-severe ARDS associated with COVID-19 and studied a treatment regimen in which supplemental oxygen was provided, followed by BiPAP/CPAP if oxygen saturation levels and respiratory rates did not improve. This was followed by IMV if signs of BiPAP failure were observed, such as worsening of dyspnea or hypoxemia, respiratory acidosis, or circulatory shock. The authors found that BiPAP was associated with a lower association with IMW therapy. Two studies showed a benefit of a stepwise approach. The recommended approach begins with HFNC as the first-line treatment. If oxygen saturation levels decrease and respiratory rates increase to >30/min, treatment should be escalated to BiPAP or CPAP. Subsequently, intubation is indicated if parameters that are used to monitor the patient’s condition worsen. These two papers suggest a strategy in which HFNC, BiPAP, and IMV are all involved at different stages of ARF depending on the status of the patient [12,89].

## 10. Conclusions

BiPAP, CPAP, and HFNC may be beneficial alternatives to IMV for COVID-19-associated ARF. One large, comprehensive RCT [56] has shown that continuous CPAP is superior to conventional oxygen therapy delivered by binasal cannula; however, HFNC did not prove superior to oxygen therapy by binasal cannula. Thus, continuous CPAP should be preferred for patients with COVID-19-associated ARF if nasal catheter oxygen therapy is insufficient for adequate oxygenation, and IMV is not indicated. HFNC therapy can be used if CPAP is not tolerated. For patients with COVID-19-associated ARF, a stepwise treatment approach that is based on patient status and includes several consecutive ventilation strategies may be the way forward.

## Figures and Tables

**Figure 1 diagnostics-11-02259-f001:**
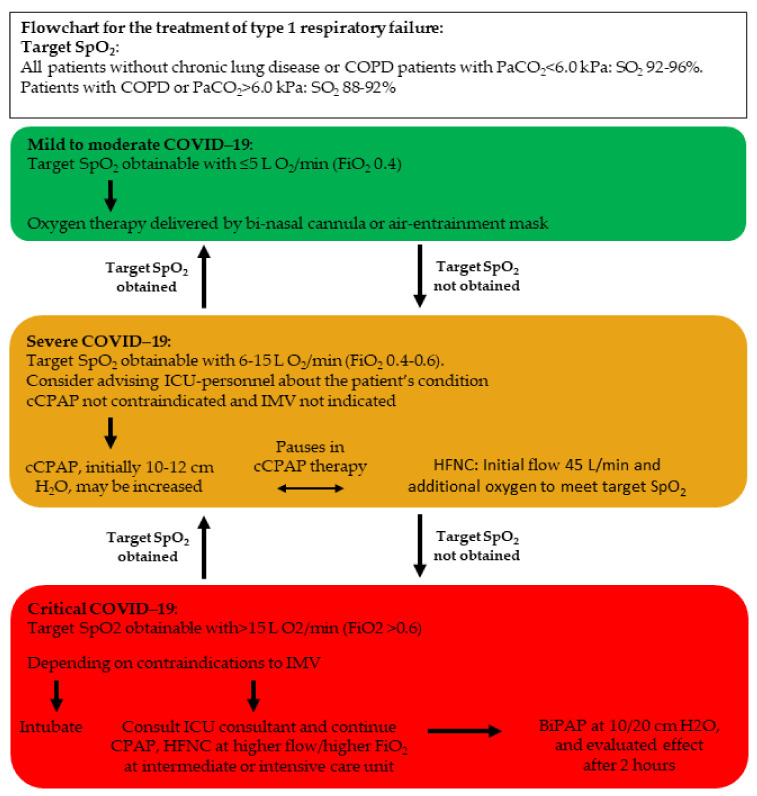
Flowchart for the treatment of type 1 respiratory failure. SpO_2_ = peripheral oxygen saturation, FiO_2_ = fraction of inspired oxygen, PaCO_2_ = partial pressure of carbon dioxide in arterial blood, cCPAP = continuous (i.e., nonintermittent) continuous positive airway pressure, HFNC = high-flow nasal cannula oxygen therapy, IMV = invasive mechanical ventilation, BiPAP = bilevel positive airway pressure, COPD = chronic obstructive pulmonary disease, COVID-19 = coronavirus disease 2019. Adapted from Nielsen Jeschke K. et al. 2020 [25].

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
