# Peer review of "Management of COVID-19-Associated Acute Respiratory Failure with Alternatives to Invasive Mechanical Ventilation: High-Flow Oxygen, Continuous Positive Airway Pressure, and Noninvasive Ventilation"

_diagnostics, 2021, doi:10.3390/diagnostics11122259_

Round 1

Reviewer 1 Report

This manuscript presented by Bonnesen et al. provides a very detailed and interesting review of current ventilatory approaches for acute respiratory failure affecting patients with COVID-19 pneumonia, but i would like to express some major concerns as follow:

  1. Subheading “Diagnosing Acute respiratory failure”:I would suggest to clarify that for a certain diagnosis of respiratory failure is mandatory performing arterial gas analysis with pO2 value less than 60 mmHg

  1. Page 3 line 122: Please better explain the value of P7F ratio in following sentence “In patients with COVID-19 associated ARF PaO2/FiO2 ratio was able to predict severity in a cohort of 421 patients [26], as well as in a smaller cohort of 150 patients, which 123 was able to calculate a sensitivity of 72% and specificity 85% [27]”

  1. As not pertinent to the context, i would suggest to remove the following sentence (Page 3 line 124) and relative reference: “PaO2/FiO2 ratio was 124 associated to pneumomediastinum in a cohort of 427 patients [28]”

  1. Part of conclusions, concerning indications and potential benefits of HFNC, are not completely in agreement with recommendations recently provided by ERJ task force (ERJ 2021 doi.org/10.1183/13993003.01574-2021) on use of HFNC on ARF patients, despite with a moderate and low certainty of evidence; therefore, I think it might be conceivable to include data from above-mentionated manuscript and to provide less strict conclusions on use of HFNC as alterative approach only when CPAP is poor tolerated

Page 3:

Figure 1 within Orange box : please specify the meaning of “Pausein”

Page 4

line 131 – please change  “ARD”  in “ARDS”;

line 171 - please change “cPAP” in “cCPAP”;

Author Response

Dear reviewer

Thank you for the opportunity to submit a revised version of our paper. We would like to thank you for your valuable comments. We have revised the paper according to these comments, and we believe it has been improved. Below are our point-by-point responses to the issues you have raised. We hope the responses and the latest revised manuscript meet with your approval.

  1. We have included the diagnostic criteria.
  2. We have elaborated the sentence with an explanation of the FiO2/PaO2 ratio and the cut-off value for the sensitivity and specificity described.
  3. The sentence has been deleted.
  4. We have included the ERS recommendations as suggested.
  5. Page 3: It reads "Pauses in", we have enlarged the text to make it more readable.
  6. Page 4 line 131 and line 171: Has been corrected.

Reviewer 2 Report

Dear authors

Thank you for your interesting work able to improve the scientific knowledges about COVID-19.

I think that this work is complete and well written.

I have the following concerns

Line 63: By contrast, a double BiPAP circuit 63 using a helmet was not associated with significant leakage.  I suggest to not specify “using an helmet” since also a full face or a oronasal mask could be used in these patients

Line 222 “BiPAP may be used to treat COVID-19 associated ARF at a pressure of 20/10 and low 222 tidal volumes such as 4–8 mL/kg”  I think that authors should specify, what does it mean “at a pressure of 20/10”?

Line 225: HRNC is “HFNC “

Line 316 One large, comprehensive RCT has shown that continuous CPAP is superior to conventional therapy; however, HFNC did not prove superior to conventional therapy. This conclusion is not clear. What is conventional therapy? What study are you citing?

Author Response

Dear reviewer

Thank you for the opportunity to submit a revised version of our paper. We would like to thank you for your valuable comments. We have revised the paper according to these comments, and we believe it has been improved. Below are our point-by-point responses to the issues you have raised. We hope the responses and the latest revised manuscript meet with your approval.

Line 63: Has been corrected.
Line 222: Has been specified.
Line 225: Has been corrected.
Line 316: Conventional therapy has been clarified and the reference for the Perkins study added.

Round 2

Reviewer 1 Report

The manuscript is acceptable in present form